# Fecal Calprotectin Is Increased in Stroke

**DOI:** 10.3390/jcm11010159

**Published:** 2021-12-29

**Authors:** Shin Young Park, Sang Pyung Lee, Woo Jin Kim

**Affiliations:** 1Department of Clinical Laboratory Science, Cheju Halla University, 38 Halladaehak-ro, Jeju-si 63092, Korea; kj901217@gmail.com; 2Brain-Neuro Center, Department of Neurosurgery, Cheju Halla General Hospital, 65 Doryeong-ro, Jeju-si 63127, Korea; nsdr745@gmail.com; 3Department of Laboratory Medicine, EONE Laboratories, 291 Harmony-ro, Yeonsu-gu, Incheon 22014, Korea

**Keywords:** stroke, gut, leukocyte L1 antigen complex, ELISA

## Abstract

Background: While there have been major advances in unveiling the mechanisms comprising the ischemic cascade of CNS, stroke continues to be a significant burden. There is a need to extend the focus toward peripheral changes, and the brain–gut axis has recently gained much attention. Our study aimed to evaluate gut inflammation and its association with blood variables in stroke using fecal calprotectin (FC). Methods: Fecal samples were obtained from 27 stroke patients and 27 control subjects. FC was quantitatively measured using a commercial ELISA. Laboratory data on the fecal sample collection were also collected, including CBC, ESR, glucose, creatinine, total protein, albumin, transaminases, and CRP. Results: There was a significant increase in FC levels in stroke patients compared to the controls. Furthermore, FC in stroke patients was negatively correlated with the Glasgow Coma Scale. Moreover, FC in stroke patients was positively correlated with CRP and negatively correlated with lymphocyte count and albumin. Conclusions: Our findings show that increased FC is associated with consciousness and systemic response in stroke and warrants further studies to elucidate the usefulness of FC in the management of stroke.

## 1. Introduction

According to a recent report from the Global Burden of Diseases, Injuries, and Risk Factors Study (GBD), stroke continues to be a significant socioeconomic burden. In 2016, stroke was the second largest cause of death globally (5.5 million) and the second most common cause of global disability-adjusted life-years (DALYs) (116.4 million), an increase from 1990 (95.3 million). There were 13.7 million new strokes in 2016 [1].

Since rodent models of focal cerebral ischemia were first reported in the early 1980s [2], there have been major advances in our knowledge regarding the complex biochemical and molecular mechanisms that comprise the ischemic cascade, and stroke mortality is gradually declining [3,4]. With more patients surviving stroke, there is a need for the focus of stroke research to turn its attention to post-stroke complications [5].

Following a stroke, gastrointestinal (GI) complications are common problems, including dysphagia, GI hemorrhage, delayed GI emptying, and colorectal dysfunction [6], and the communication between the central nervous system (CNS) and the gut in animal models has recently gained attention in stroke research [7]. While fecal calprotectin (FC) is a relatively new marker not yet explored in stroke research, it has been a useful marker of gastrointestinal inflammation predominantly released by neutrophils in the gut [8].

Here, in a pilot prospective observational study, we aimed to investigate FC alterations in stroke patients compared with controls and explore gastrointestinal inflammation in stroke patients. The second aim was to investigate the correlation with monitoring markers used in stroke management such as the Glasgow Coma Scale (GCS), C-reactive protein (CRP), complete blood count (CBC), and kidney battery, and to explore the usefulness of FC in stroke monitoring.

## 2. Materials and Methods

### 2.1. Subjects

In the present study, fecal samples were obtained from 54 subjects: 27 stroke patients (STR) and 27 healthy control subjects (CON). Stroke patients, including both hemorrhagic and ischemic strokes, were directly admitted from the emergency departments (ER) to intensive care units (ICU). The subjects were enrolled from September 2018 to April 2019. Written informed consent was obtained from the subjects or their guardians. Diagnoses were based on computed tomography (CT) scans or magnetic resonance imaging (MRI) studies. The patients had no history of colorectal or systemic inflammatory conditions. The study was approved by the ethical committee of Cheju Halla University (IRB approval number: 1044348-20180713-HR-007-01). 

### 2.2. Demographic Characteristics and Laboratory Data

Baseline demographic and clinical characteristics of the stroke patients were collected (Table 1 and Table 2, respectively). These included patient sex, age, body mass index (BMI), stroke type, and comorbidities. Laboratory data of fecal samples were also collected, including complete and differential blood counts using an XN-10 automated hematology analyzer (Sysmex, Kobe, Japan), erythrocyte sedimentation rate (ESR) using a TEST 1 automated analyzer (Alifax, Padova, Italy), and serum levels of glucose, creatinine, total protein (TP), albumin, aspartate transaminase (AST), alanine transaminase (ALT), and C-reactive protein (CRP) using a Cobas 8000 c702 automated analyzer (Roche Diagnostics, Basel, Switzerland). The laboratory characteristics are summarized in Table 3. 

### 2.3. FC Quantitation

Subjects provided a single fecal sample for calprotectin measurement as early as possible during their ICU stay. Experimental procedures followed the methods reported by Park et al. [9]. Fecal samples were kept frozen on receipt at −80 °C. On the days of sample preparation, the frozen fecal samples were thawed at room temperature (RT) for 30 min. Thereafter, a single 100 mg aliquot was suspended in 1 mL of fecal extraction buffer consisting of 0.1 M Tris-buffered saline with Tween 20, pH 8.0 (Kisanbio Tech, Seoul, Korea) with 0.5% bovine serum albumin and 10 mM CaCl_2_. The sample was homogenized for 5 min with a vortex mixer (Scientific Industries, Bohemia, NY, USA). The homogenates were centrifuged for 15 min at 10,000× *g* at RT. The top portions of the supernatants were removed and kept at −80 °C until an enzyme-linked immunosorbent assay (ELISA) was carried out. 

FC was quantitatively measured using a Legend MaxHuman MRP8/14 (Calprotectin) ELISA Kit (BioLegend, San Diego, CA, USA). The frozen fecal extracts were defrosted and diluted in a 1:1000 assay buffer. Standards and diluted samples (50 µL) were added to anti-human MRP8/14 pre-coated ELISA plates, which were then shielded and incubated at RT for 1 h while agitating at 200 rpm. Human MRP8/14 detection antibody solution (100 µL) was added to each well after washing the wells four times with washing buffer. The plates were shielded and incubated at RT for 30 min while agitating. After incubation with the detection antibody solution, the wells were washed five times with wash buffer, and 100 µL of substrate solution was added. Stop solution (100 µL) was added, and the absorbance was read at 450 and 570 nm using a PowerWave XS2 Microplate Spectrophotometer (BioTek, Winooski, VT, USA). The readings at 570 nm were subtracted from those at 450 nm. Calprotectin values were expressed as micrograms per gram of fecal sample. 

### 2.4. Statistical Analysis

Per group, 25 subjects were needed to achieve a power of 90% with two-sided 5% significance and 0.2 for an effective size, according to Whitehead et al. [10]. Data were analyzed using the GraphPad Prism statistical software package (GraphPad Software Inc., La Jolla, CA, USA). The Student’s t-test or Fisher’s exact test were used according to variables to compare demographic characteristics of the study population. Calprotectin values and blood variables were presented as the mean ± SD. Mann–Whitney tests were performed to compare FC concentrations between the stroke patients and the control group. Pearson or Spearman correlation analyses were conducted to determine the association between FC and laboratory data in the stroke patient group. All tests were two-sided, and *p* < 0.05 was considered statistically significant. A normality test was conducted using the Kolmogorov–Smirnov (KS) normality test.

## 3. Results

### 3.1. FC Concentrations in Healthy Controls and Stroke Patients

Mean FC concentration for the healthy CON (*n* = 27) was 23.0 µg/g (range 0.0–166.1 µg/g). In comparison, the mean FC concentration for STR patients (*n* = 27) was 335.3 µg/g (range 0.0–1206.5 µg/g). The Mann–Whitney test showed there was a statistically significant difference between the FC concentrations of the CON and STR groups (Table 4). 

### 3.2. Association among FC Concentration and GCS and Blood Variables in Stroke

Mean levels of blood variables in STR were 10,303/µL white blood cells (WBC), 7759/µL neutrophils, 1378/µL lymphocytes, 235,000/µL platelets, 4.6 mg/dL CRP, 11.3 g/dL hemoglobin, 24.3 mm/h ESR, 46.4 U/L AST, 41.0 U/L ALT, 6.1 g/dLTP, 3.5 g/dL albumin, 151.4 mg/dL glucose, and 0.6 mg/dL creatinine. Mean levels of GCS were 9.8 (Table 3). The blood variables with a significant correlation with GCS included lymphocyte count (positive), CRP (inverse), albumin (positive), and ESR (inverse). GCS showed a negative correlation with FC. The blood variables with a significant correlation with FC included lymphocyte count (inverse), CRP (positive), albumin (inverse), hemoglobin (inverse), and total protein (inverse) (Table 5).

## 4. Discussion

We studied calprotectin levels in fecal samples of stroke patients and showed that FC was significantly increased in stroke patients. According to the available literature on the determinants of FC levels, associated drugs with FC levels include proton pump inhibitors (PPI) (elevation), non-steroidal anti-inflammatory drugs (NSAID) (elevation) [11], or antibiotics (depression) [12]. The patients in this study took neither PPI nor NSAI but did take intravenous antibiotics around the collection time of fecal samples. Considering the available literature and medication profiles of patients, it seems less likely that the medication affected the increase in FC. 

Calprotectin is a small acidic protein isolated from leukocytes in 1983 [13], which is involved in inflammatory processes. Since the late 1980s, when calprotectin was described from stool extracts using ELISA [14], related research has increased every year. Numerous studies have shown that FC is significantly elevated in patients with IBD and such increased levels correlate well with both endoscopic and histologic assessment of disease activity. Lee et al. recently reported that a cutoff value of 201.3 μg/g for FC predicted endoscopic inflammation in the ulcerative colitis with a sensitivity of 81.8% and a specificity of 100% [15]. Gastrointestinal complications after stroke are common and are often associated with an increased hospital stay, development of further complications, and even increased mortality [16]. The mean FC of stroke patients in our study was higher than 201.3 μg/g. Furthermore, objective non-invasive markers of gastrointestinal inflammation in stroke are rarely found. Therefore, furtherer investigation is warranted to find the role of FC in detecting gastrointestinal inflammation in patients with acute stroke.

Our correlation study showed that FC had a positive association with CRP and a negative association with lymphocytes and albumin. On the other hand, GCS had a negative association with CRP and a positive association with lymphocytes and albumin. These findings are consistent with the finding that FC is negatively associated with GCS. Many studies have been conducted regarding the association between those markers and acute stroke. CRP is a marker of inflammation, and its elevation in acute stroke is reportedly a poor prognostic factor [17]. GCS is widely used in assessing level of consciousness, and low levels of GCS are reportedly associated with higher mortality [18]. Lymphocytes are a subset of leukocytes, and their depletion in acute stroke is associated with poor neurologic outcome [19]. Albumin serves as a biochemical marker of nutritional status, and hypoalbuminemia is associated with poor functional outcome [20]. In an IBD study, Lee et al. showed that FC had a positive correlation with CRP levels (*r* = 0.379, *p* < 0.001) and a negative correlation with serum albumin levels (*r* = −0.426, *p* < 0.001) in patients with ulcerative colitis [15]. Other studies showed that low serum albumin and high CRP levels are associated with severe disease activity in patients with ulcerative colitis [21]. Furthermore, a correlation was found between fecal calprotectin and clinical outcome in patients with ulcerative colitis [22]. These findings warrant further study to find the role of FC as a prognostic marker in patients with acute stroke. 

The brain controls our body through complex neurohumoral mechanisms. Severe cerebral insults, such as acute stroke, can induce changes in neurosensory or neuromotor pathways, thereby enhancing systemic immune response and causing secondary peripheral organ damage [23,24]. To evaluate and monitor acute stroke, clinical scales, such as GCS and the aforementioned laboratory tests, are performed at the patient’s bedside. Though FC is rarely ordered in acute stroke patients, it has been used as one of the most reliable, non-invasive tools for managing inflammatory bowel disease (IBD) in clinical practice [25]. Here, we explored the method of measuring fecal calprotectin and its association with patient variables in stroke. We found that FC is significantly elevated in patients with stroke and is correlated with laboratory variables, which are performed to evaluate and monitor acute stroke in the clinical field. Our findings may help clinicians be more attentive to gut response in parallel with systemic response in stroke and warrant further studies to elucidate the usefulness of FC in stroke management.

## Figures and Tables

**Table 1 jcm-11-00159-t001:** Demographic characteristics of the study population.

	CON (*n* = 27)	STR (*n* = 27)	*p*
Age (year)	59.4 ± 14.5	60.4 ± 15.2	0.533
Sex (M/F)	16/11	15/12	1
BMI (kg/m^2^)	23.4 ± 3.4	24.0 ± 3.5	0.661
Comorbidities			
DM (number)	5	5	1
Hypertension (number)	8	14	0.165
CAD (number)	0	2	0.490
Medication			
PPI (number)	0	0	NA
NSAID (number)	0	0	NA
ABX (number)	0	27	<0.0001

Abbreviations: CON, healthy controls; STR, stroke patients; BMI, body mass index; DM, diabetes mellitus; CAD, coronary heart disease; PPI, proton pump inhibitor; NSAID, non-steroidal anti-inflammatory drug; ABX, antibiotics. Values expressed as mean ±  SD.

**Table 2 jcm-11-00159-t002:** Clinical characteristics of the stroke patients (*n* = 27).

Subject ID	Age	Sex	Stroke Type	Smoking	Comorbid	GCS on Admission	mRS at 3 Months	Adm Date (Mo/Da/Yr)	FC Date (Mo/Da/Yr)	FC Conc (µg/g)	PPI	NSA	AB
STR01	60	M	Ischemic	yes	no	8	3	09/03/18	09/10/18	220.2	no	no	yes
STR02	48	M	Hemorrhagic	no	no	4	5	09/07/18	09/10/18	86.6	no	no	yes
STR03	59	F	Hemorrhagic	no	DM	6	3	09/14/18	09/17/18	1004.3	no	no	yes
STR04	54	M	Ischemic	yes	no	8	3	09/29/18	10/08/18	1206.5	no	no	yes
STR05	58	M	Hemorrhagic	yes	no	3	4	11/08/18	11/14/18	507.9	no	no	yes
STR06	77	F	Hemorrhagic	no	HTN	14	3	02/11/19	02/26/19	106.3	no	no	yes
STR07	81	F	Hemorrhagic	no	no	7	5	02/13/19	03/05/19	57.2	no	no	yes
STR08	41	M	Hemorrhagic	yes	no	14	1	03/04/19	03/13/19	216.0	no	no	yes
STR09	41	F	Hemorrhagic	no	no	14	3	02/24/19	03/17/19	52.9	no	no	yes
STR10	68	F	Ischemic	no	DM, HTN	15	6	03/07/19	04/29/19	34.4	no	no	yes
STR11	39	M	Hemorrhagic	yes	no	14	3	03/18/19	03/28/19	14.5	no	no	yes
STR12	48	F	Hemorrhagic	no	HTN	14	3	04/02/19	04/09/19	106.8	no	no	yes
STR13	70	M	Ischemic	yes	DM, HTN	10	3	04/13/19	04/19/19	804.3	no	no	yes
STR14	51	M	Hemorrhagic	yes	HTN	14	3	04/13/19	04/23/19	13.3	no	no	yes
STR15	77	M	Ischemic	no	DN, HTN, CAD	6	5	08/17/18	09/04/18	899.4	no	no	yes
STR16	43	M	Hemorrhagic	no	no	7	3	10/21/18	10/25/18	195.1	no	no	yes
STR17	63	M	Hemorrhagic	no	HTN	7	4	10/13/18	10/29/18	744.2	no	no	yes
STR18	43	M	Hemorrhagic	no	DM	7	3	10/28/18	11/15/18	18.0	no	no	yes
STR19	66	F	Hemorrhagic	no	HTN	7	3	01/27/19	02/27/19	633.3	no	no	yes
STR20	71	F	Hemorrhagic	no	HTN	15	2	02/12/19	02/18/19	0.0	no	no	yes
STR21	86	F	Hemorrhagic	no	no	7	4	02/10/19	02/18/19	592.6	no	no	yes
STR22	84	F	Hemorrhagic	no	HTN	4	5	03/08/19	03/15/19	230.9	no	no	yes
STR23	52	M	Ischemic	no	HTN	9	3	03/10/19	03/15/19	233.9	no	no	yes
STR24	81	F	Ischemic	no	HTN	14	6	03/14/19	03/15/19	65.5	no	no	yes
STR25	79	F	Hemorrhagic	no	HTN	13	3	03/30/19	04/03/19	0.0	no	no	yes
STR26	51	M	Ischemic	yes	CAD	12	3	03/23/19	04/08/19	872.1	no	no	yes
STR27	41	M	Hemorrhagic	yes	HTN	12	3	04/01/19	04/09/19	134.7	no	no	yes

Abbreviations: STR, stroke patients; DM, diabetes mellitus; HTN, hypertension; CAD, coronary heart disease; GCS, Glasgow Coma Scale; mRS, modified Rankin scale; Adm date, date of admission; FC, fecal calprotectin test; Conc, concentration; PPI, proton pump inhibitors; NSA, non-steroidal anti-inflammatory drugs; AB, antibiotics.

**Table 3 jcm-11-00159-t003:** Laboratory characteristics and GCS of the stroke patients.

Variable	Result (*n* = 27)	Reference Range	KS Normality Test Alpha = 0.05
FC (µg/g)	335.3 ± 367.6	<50	No
WBC (/µL)	10,303 ± 2737	4000–11,000	Yes
Neutrophil (/µL)	7759 ± 2737	1500–7000	Yes
Lymphocyte (/µL)	1378 ± 706.4	1500–4500	Yes
Hb (g/dL)	11.3 ± 1.4	12–16	Yes
Platelet (/µL)	235,000 ± 102,369	150,000–450,000	Yes
CRP (mg/dL)	4.6 ± 3.7	<0.3	No
ESR (mm/hr)	24.3 ± 22.1	<20	Yes
AST (U/L)	46.4 ± 65.3	10–40	No
ALT (U/L)	41.0 ± 53.1	10–40	No
TP (g/dL)	6.1 ± 0.7	6–8	Yes
Alb (g/dL)	3.5 ± 0.7	3.5–5	No
Glucose (mg/dL)	151.4 ± 50.2	60–110	No
Creatinine (mg/dL)	0.6 ± 0.1	0.5–1.2	Yes
GCS	9.8 ± 3.8	15	No

Abbreviation: FC, fecal calprotectin; WBC, white blood cell; Hb, hemoglobin; CRP, C-reactive protein; ESR, erythrocyte sedimentation rate; AST, aspartate transaminase; ALT, alanine transaminase; TP, total protein; Alb, albumin; GCS, Glasgow Coma Scale; KS, Kolmogorov–Smirnov. The values are expressed as the mean ± SD.

**Table 4 jcm-11-00159-t004:** Fecal calprotectin concentrations of healthy controls and stroke patients.

	CON (*n* = 27)	KS Normality Test	STR (T1) (*n* = 27)	KS Normality Test	*p*
FC (µg/g)	23.0 ± 39.7	no	335.3 ± 367.6	no	<0.0001

Abbreviations: FC, fecal calprotectin; CON, healthy controls; STR, stroke patients; KS, Kolmogorov–Smirnov. The values are expressed as the mean ± SD.

**Table 5 jcm-11-00159-t005:** Correlations among FC, GCS, and blood variables in stroke patients.

Variable	Variable	Variable	Analysis	r	*p*
GCS	Lymphocyte		Pearson	0.4187	0.0297
	CRP		Spearman	−0.65402	0.0002
	Albumin		Spearman	0.5636	0.0022
	ESR		Pearson	−0.4073	0.0350
GCS		FC	Spearman	−0.5567	0.0026
	Lymphocyte	FC	Pearson	−0.4074	0.0349
	CRP		Spearman	0.5768	0.0016
	Albumin		Spearman	−0.4558	0.0169
	Hemoglobin		Pearson	−0.4403	0.0215
	Total protein		Pearson	−0.5553	0.0026

Abbreviations: GCS, Glasgow Coma Scale; CRP, C-reactive protein; FC, fecal calprotectin.

## Data Availability

Data supporting the reported result can be accessed by corresponding with the authors.

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
