# Peer review of "Fecal Calprotectin Is Increased in Stroke"

_jcm, 2021, doi:10.3390/jcm11010159_

Round 1

Reviewer 1 Report

Authors answers to previous comments: "Though further studies are needed, FC in stroke patients may be increased to defend host and prevent opportunistic infection against indigenous gut microbiota in such challenging situation as gut dysfunction."

IMHO it would really improve the quality of paper if authors add the to discussion, why the FC is so important to stoke patients - because of the complexity of brain-gut axis (microbiota, inflammatory cytokines, hormones ect.). Also what is necessary to do in further studies to confirm the proposed hypotheses.

Author Response

  • We appreciate for your precious comments. Following your instructions, we revised our manuscript so that the aims and importance of our study in stroke patients would be more clear. The descriptions are detailed in introduction and discussion.

While Fecal calprotectin (FC) is a relatively new marker and never explored in stroke research, it has been a useful marker of gastrointestinal inflammation that is released by predominantly neutrophils in the gut [8]. This is a pilot prospective observational study and the first aim was to investigate alteration in FC in stroke patients in comparison with those in control group and to explore gastrointestinal inflammation in stroke patients. The second aim was to investigate correlation with such monitoring markers used in stroke management as Glasgow Coma Scales (GCS), C-reactive protein (CRP), Complete blood count (CBC), kidney battery and to explore usefulness of FC in stroke monitoring.

The facts that mean FC of stroke patients in our study were higher than 201.3 μg/g and that objective noninvasive markers of gastrointestinal inflammation in stroke are hardly found warrant larger study to find the role of FC as detecting gastrointestinal inflammation in patients with acute stroke.

Our correlation study showed that FC had positive association with CRP while it had negative association with lymphocyte and albumin. On the other hand, GCS had negative association with CRP while it had positive association with lymphocyte and albumin. These findings are consistent with the finding that FC had negative association with GCS. Many studies have been conducted regarding the association between those markers and acute stroke. In IBD study, Lee et al showed that FC had positive correlation with CRP levels (r = 0.379, p < 0.001) and negative correlation with serum albumin levels (r = −0.426, p < 0.001) in patients with ulcerative colitis [21]. Other researchers showed that low serum albumin and high CRP levels were associated with severe disease activity in patients with ulcerative colitis[22] and correlation between fecal calprotectin and clinical outcome in patients with ulcerative colitis [23]. These findings warrant larger study to find the role of FC as prognostic marker in patients with acute stroke.

Reviewer 2 Report

Dear author:

Major comments are shown as following:   

  1. The study aim is not clear and the data could not fully support the conclusion.
  2. The calculation of sample sizes in this study is not clear.
  3. The role of FC in stoke patient has not been well reviewed and summarized in the introduction part.
  4. The demographic comparison between patient with stroke and health control is inadequate. In table 1, only three parameters were listed as age, sex and BMI. We don`t known other important factors that might associate with the presentation of FC, such as comorbidity, medication, diet status.....
  5. Table 2: FC were checked in the acute stage (within 7 days from admission). As you mention in the second and third part of introduction, FC is a markers to show the CNS and gut communication in the post-acute phase. I did not see any data addressed on this part. 
  6. Table 4 showed the difference of FC level between CON and STR group. Table 5 showed the negative correlation between FC & GCS, and FC & lymphocyte in STR group. I don`t really understand the clinical meaning and application of FC in stroke patient. Do you suggest we should check FC level to replace the GCS score?   

Other comments:

  1.  The manuscript should be revised by a native English speaker.
  2.  Introduction should focus on the association of FC and stroke.
  3. Methods: The effective sample size should be presented.
  4. Table: need to re-organized
  5. Abstract: not the stand form of MDPI abstract. 

Author Response

We appreciate for your precious comments and our replies as follows:

Major comments:   

  1. The study aim is not clear and the data could not fully support the conclusion.

--> The first aim was to investigate alteration in FC in stroke patients in comparison with those in control group and to explore gastrointestinal inflammation in stroke patients. The second aim was to investigate correlation with such monitoring markers used in stroke management as Glasgow Coma Scales (GCS), C-reactive protein (CRP), Complete blood count (CBC), kidney battery and to explore usefulness of FC in stroke monitoring.

  1. The calculation of sample sizes in this study is not clear 

--> Our study is a pilot prospective observational study and Per group, 25 subjects were needed to achieve a power of 90% with two-sided 5% significance and 0.2 for effective size according to Whitehead et al [10]. 

  1. The role of FC in stoke patient has not been well reviewed and summarized in the introduction part.

--> While Fecal calprotectin (FC) is a relatively new marker and never explored in stroke research, it has been a useful marker of gastrointestinal inflammation that is released by predominantly neutrophils in the gut. Calprotectin is a small acidic protein involved in inflammatory process and was isolated from leukocytes in 1983. Since the late 1980s when calprotectin could be determined in stool extracts using ELISA, related research is increasing every year. Numerous studies have shown that FC is significantly elevated in patients with IBD and such increased levels correlate well with both endoscopic and histologic assessment of disease activity.

  1. The demographic comparison between patient with stroke and health control is inadequate. In table 1, only three parameters were listed as age, sex and BMI. We don`t known other important factors that might associate with the presentation of FC, such as comorbidity, medication, diet status.....

--> More parameters were added to table 1 except diet status. Regarding diet status, please excuse our ignorance. We wonder how to compare two groups about diet status... 

  1. Table 2: FC were checked in the acute stage (within 7 days from admission). As you mention in the second and third part of introduction, FC is a markers to show the CNS and gut communication in the post-acute phase. I did not see any data addressed on this part. 

--> Table 2 shows Adm Date and FC date. According to patients' status, the intervals from admission to first FC test are variable from 3 days to 52 days. And We are sorry that we cannot find 'gut communication in the post-acute phase' in introduction. 

  1. Table 4 showed the difference of FC level between CON and STR group. Table 5 showed the negative correlation between FC & GCS, and FC & lymphocyte in STR group. I don`t really understand the clinical meaning and application of FC in stroke patient. Do you suggest we should check FC level to replace the GCS score?   

--> Our correlation study showed that FC had positive association with CRP while it had negative association with lymphocyte and albumin. On the other hand, GCS had negative association with CRP while it had positive association with lymphocyte and albumin. These findings are consistent with the finding that FC had negative association with GCS. We suggest check FC level for outcome or monitoring rather than to replace GCS score. Please see the submitted pdf.

Other comments:

  1.  The manuscript should be revised by a native English speaker.
  2.  Introduction should focus on the association of FC and stroke.
  3. Methods: The effective sample size should be presented.
  4. Table: need to re-organized
  5. Abstract: not the stand form of MDPI abstract. 

Round 2

Reviewer 2 Report

All questions has been answered and response well. 

Suggest accept for publication.

This manuscript is a resubmission of an earlier submission. The following is a list of the peer review reports and author responses from that submission.

Round 1

Reviewer 1 Report

Authors found that FC is elevated in patients with stroke, what confirms the role of brain-gut or gut-brain axis.

IMHO the authors could add to discussion how the study results can improve current management of patient with stroke.

Reviewer 2 Report

In their study, Park et al. aimed to elucidate the correlation between fecal calprotectin (FC) and serum parameters (albumin, C-reactive protein (CRP) and neutrophil-to-lymphocyte ratio (NLR)) in human stroke. The impact of this study is clearly limited due to the low clinical relevance, highly observational nature of the experiment as well as poor statistical methodology to verify their results. The study in its present form leaves several questions and the manuscript should not be accepted for publication in the Journal of Clinical Medicine. In that context, questions and suggestions for improving this study are detailed below:

The study is highly descriptive and there is a lack of medical relevance. Do the authors see the potential of FC as a prognostic marker? Is there any knowledge on the long-term outcome based on FC levels or on the course of FC levels before/after stroke? How can the determination of FC levels be of importance after experiencing a stroke?

Before correlating FC levels to albumin/CRP and NLR, did the authors correct their data for all other parameters? It should be excluded that the FC levels were, for instance, changed due to post-stroke medications or any pre-existing medical conditions. However, since it would be unethical to have a stroke control group which experienced a stroke but didn’t receive any medication, the possibility that FC levels are changed due to the stroke alone and not due to the drugs, cannot be ruled out.

The introduction is too short and since the study is based on human samples, the authors should not start their introduction by mentioning rodent disease models.